# BCDnet: Parallel heterogeneous eight-class classification model of breast pathology

**Qingfang He** *, **Guang Cheng, Huimin Ju**

Institute of Computer Technology, Beijing Union University, Beijing, China

* qingfang@buu.edu.cn

## Abstract

Breast cancer is the cancer with the highest incidence of malignant tumors in women, which seriously endangers women's health. With the help of computer vision technology, it has important application value to automatically classify pathological tissue images to assist doctors in rapid and accurate diagnosis. Breast pathological tissue images have complex and diverse characteristics, and the medical data set of breast pathological tissue images is small, which makes it difficult to automatically classify breast pathological tissues. In recent years, most of the researches have focused on the simple binary classification of benign and malignant, which cannot meet the actual needs for classification of pathological tissues. Therefore, based on deep convolutional neural network, model ensembleing, transfer learning, feature fusion technology, this paper designs an eight-class classification breast pathology diagnosis model BCDnet. A user inputs the patient's breast pathological tissue image, and the model can automatically determine what the disease is (Adenosis, Fibroadenoma, Tubular Adenoma, Phyllodes Tumor, Ductal Carcinoma, Lobular Carcinoma, Mucinous Carcinoma or Papillary Carcinoma). The model uses the VGG16 convolution base and Resnet50 convolution base as the parallel convolution base of the model. Two convolutional bases (VGG16 convolutional base and Resnet50 convolutional base) obtain breast tissue image features from different fields of view. After the information output by the fully connected layer of the two convolutional bases is fused, it is classified and output by the Soft-Max function. The model experiment uses the publicly available BreaKHis data set. The number of samples of each class in the data set is extremely unevenly distributed. Compared with the binary classification, the number of samples in each class of the eight-class classification is also smaller. Therefore, the image segmentation method is used to expand the data set and the non-repeated random cropping method is used to balance the data set. Based on the balanced data set and the unbalanced data set, the BCDnet model, the pre-trained model Resnet50+ fine-tuning, and the pre-trained model VGG16+ fine-tuning are used for multiple comparison experiments. In the comparison experiment, the BCDnet model performed outstandingly, and the correct recognition rate of the eight-class classification model is higher than 98%. The results show that the model proposed in this paper and the method of improving the data set are reasonable and effective.

**Data Availability Statement:** The source code, detailed model structure diagram and other data involved in this article have been uploaded to https://github.com/yeaso/bcdnet.

**Funding:** The work is supported by Beijing Natural Science Foundation(No.L191006) and Academic Research Projects of Beijing Union University (No. XP202021).

**Competing interests:** The authors have declared that no competing interests exist.

## Introduction

Breast cancer is a common malignant tumor in clinic. According to the "2018 Global Cancer Statistics" report published by the official journal of the American Cancer Society, breast cancer is the cancer with the highest incidence of malignant tumors in women [1], and its incidence is increasing year by year and younger [2]. At present, the preferred treatment for breast cancer is surgery [3]. Early detection and early treatment are the key to improving the survival rate of breast cancer patients. Diagnosis based on pathological tissue sections is the gold standard for diagnosing breast cancer. Doctors make a diagnosis from the patient's pathological tissue slice images, which not only requires professional experience, but also time-consuming and laborious, and the diagnosis results are affected by subjective human factors. Therefore, a small number of patients will be under-treated or over-treated due to differences in the diagnosis and treatment of doctors [4]. With the help of computer vision technology, the diagnosis and treatment equipment automatically makes a diagnosis from the patient's pathological tissue slice image, which can not only improve the diagnosis efficiency, but also assist the doctor to provide more objective and accurate diagnosis results.

With the rapid development of computer software and hardware technology, especially the wide application of convolutional neural networks (CNN) based on large data sets in the fields of natural language processing, object recognition, image classification and recognition [5–8], this laid the foundation for the application of CNN in the classification of breast cancer pathological images [9]. Since 2015, Spanhol et al. published the BreaKHis breast cancer pathology image data set [10]. Based on this data set, a series of research results have been achieved in breast cancer recognition using convolutional neural networks. Paper [10] carried out a study on the two classification of pathological images of breast cancer. They used different classification algorithms such as Local Binary Pattern (LBP), Gray Level Co-occurrence Matrix (GLCM), Support Vector Machine, Random Forest, etc for classification experiments. The published research results of them showed that the accuracy of breast cancer malignant recognition is 80%-85%. Bayramoglu et al. [11] constructed single-task and multi-task convolutional neural networks for binary classification research, which predicted that the recognition rate of malignant tumors was about 83%,concluded that the recognition rate has nothing to do with the magnification of the image. Bayramoglu et al. combined with DeCaf (A Deep Convolutional Activation Feature for Generic Visual Recognition) feature evaluation method to adjust the CNN architecture and achieved 86.3% accuracy on breast cancer pathological images [11]. Spanhol et al. [12] used the classic network structure AlexNet on the BreaKHis data set to conduct a binary classification study, and the recognition accuracy reached 85% to 90%. In 2020, Mesut et al. [13] integrated the convolutional neural network model and the autoencoder network model, and the best recognition rate for binary classification was 98.59%. AMT, BKBZ, CBE, et al. [14] conducted a four-class classification experiment based on the publicly available BreaKHis data set. Their model used the attention mechanism and residual module structure, and used enhanced data in experiments. The four-class classification model they proposed achieved a 98.80% recognition rate. Yao, H, et al. [15] proposed a parallel fusion network structure model composed of CNN and RNN. Their model achieved a 97.5% recognition rate for four-class classification of breast cancer pathological tissues.

Judging from the research results released above, the research on automatic identification of breast pathological tissues is moving from binary classification to multi-class classification, and the learning methods used are changing from ordinary convolutional neural networks, deep neural networks, and complex deep neural networks to network convergence.

Pathological tissue image classification is different from traditional image classification (such as the recognition of cats and dogs) in image characteristics and data set size.

Pathological tissue images have the characteristics of differential blur, feature diversity, cell overlap phenomenon, uneven color distribution, etc., especially pathological tissue images have the characteristics of small data set size and uneven number of benign and malignant samples, which affects the classification performance of the model, especially it brings challenges to multi-class classification research. From the relevant Paper published in recent years, simple binary classification of benign and malignant has reached a high level. Currently, there are few published research results of multi-class classification, and the published research results of multi-class classification are generally four-class classification. Four-class classification cannot meet the actual clinical diagnosis needs. Therefore, this article proposes an eight-class classification model. It is expected that the model will map the given breast pathological tissue images in adenopathy (A), fibroadenoma (F), tubular adenoma (TA) and phyllodes tumor (PT), Make correct judgments in ductal carcinoma (DC), lobular carcinoma (LC), mucinous carcinoma (MC) and papillary carcinoma (PC)).

The main contributions of this article are:

i. An eight-class classification parallel heterogeneous fusion network model BCDnet is proposed. The networks VGG16 [16] and Resnet50 [17] with excellent performance and different structures are used as the convbase of the BCDnet model, so that the model has the characteristics of "strong combination, selection of the best".

ii. According to the analysis of multiple sets of experimental results, if there is a serious imbalance in the number distribution of the sample data in each category, by using the random non-repetitive segmentation method on the original image, increasing the number of small samples to balance the number of samples in the data set, which can significantly improve the overall model performance.

iii. In model training, there are always some samples that are difficult to cluster, and the research direction is given for this. When the model performance reaches a higher level, if you want to further improve the model, this article believes that how to improve the data set [18], how to carry out algorithm research and technical improvement on those samples that are difficult to identify, should be the best choice.

## Method

### Data set preparation

This article uses the public data set BreaKHis [10], which was collected by the Brazilian P&D laboratory through clinical research. All samples in this dataset are from 82 patients, and this dataset contains microbiopsy images of benign and malignant breast tumors. The samples were taken from breast tissue biopsy slides and stained with hematoxylin and eosin (HE). The samples were collected by surgical (opened) biopsy (SOB) and marked by pathologists in the P&D laboratory. The images were obtained by using magnification factors of 40X, 100X, 200X, 400X in a 3-channel RGB true color space, and the image size is 700x460. The data set contains a benign subset and a malignant subset. The benign subset contains four subcategories: adenopathy (A), fibroadenoma (F), tubular adenoma (TA), and phyllodes tumor (PT). The benign subset has a total of 2480 samples. The malignant subset contains four subcategories: ductal carcinoma (DC), lobular carcinoma (LC), mucinous carcinoma (MC) and papillary carcinoma (PC). There are 5429 samples in the malignant subset. Table 1 shows the specific distribution of each sub-category, number of people, and corresponding sample number of the BreaKHis data set.

**Table 1. Overview of BreaKHis data set.**

| Magnification | Benign | | | | Benign | Malignant | | | | Malignant | Total |
|---|---|---|---|---|---|---|---|---|---|---|---|
| | A | F | PT | TA | Total | DC | LC | MC | PC | Total | |
| 40X | 114 | 253 | 109 | 149 | 625 | 864 | 156 | 205 | 145 | 1370 | 1995 |
| 100X | 113 | 260 | 121 | 150 | 644 | 903 | 170 | 222 | 142 | 1437 | 2081 |
| 200X | 111 | 264 | 108 | 140 | 623 | 896 | 163 | 196 | 135 | 1390 | 2013 |
| 400X | 106 | 237 | 115 | 130 | 588 | 788 | 137 | 169 | 138 | 1232 | 1820 |
| Patients | 4 | 10 | 3 | 7 | 24 | 38 | 5 | 9 | 6 | 58 | 82 |

It can be seen from Table 1 that the number of pathological tissue image samples from 82 patients is very unevenly distributed in the eight categories, and the number of samples corresponding to the four magnification ratios is not one-to-one. According to the sample statistical files provided by the data set, the number of pathological tissue pictures of each patient in the data set is different, even if the number of pathological tissue pictures of the same patient is different at different magnifications. As shown in Fig 1, the number of pathological tissue pictures of 38 DC patients provided for the BreaKHis data set at four different magnifications.

From the values in the Benign Total and Malignant Total columns in Table 1, it can be seen that the ratio of the number of benign samples to the number of malignant samples is about

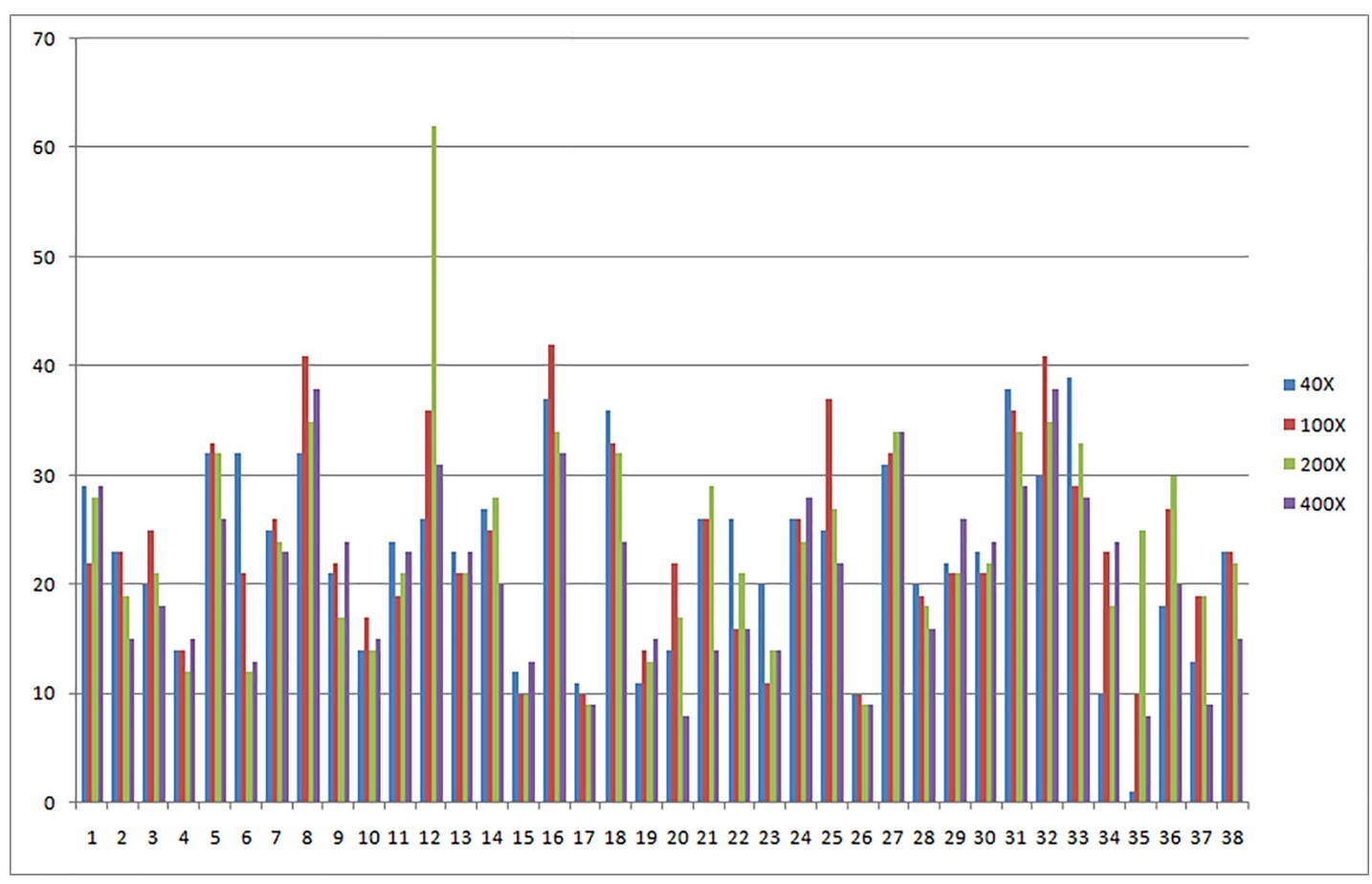

**Fig 1. Statistics of pathological tissue pictures of 38 DC patients in BreaKHis dataset.**

**Table 2. The distribution of various samples after the image segmentation method is used to expand the data set.**

| Magnification | Benign | | | | Benign | Malignant | | | | Malignant | Total |
|---|---|---|---|---|---|---|---|---|---|---|---|
| | A | F | PT | TA | Total | DC | LC | MC | PC | Total | |
| 40X | 684 | 1518 | 654 | 894 | 3750 | 5184 | 936 | 1230 | 870 | 8220 | 11970 |
| 100X | 678 | 1560 | 726 | 900 | 3864 | 5418 | 1020 | 1332 | 852 | 8622 | 12486 |
| 200X | 666 | 1584 | 648 | 840 | 3738 | 5376 | 978 | 1176 | 810 | 8340 | 12078 |
| 400X | 636 | 1422 | 690 | 780 | 3528 | 4728 | 822 | 1014 | 828 | 7392 | 10920 |
| Patients | 4 | 10 | 3 | 7 | 24 | 38 | 5 | 9 | 6 | 58 | 82 |

1:2. From the values in the columns A, F, PT, TA, DC, LC, MC, and PC in Table 1, the ratio of the minimum number of samples to the maximum number of samples is about 1:8. It can be seen that the distribution of the number of samples in the eight categories of the data set is extremely uneven. In order to ensure the training effect of the design model and the generalization performance of the model, the following data set processing strategies are formulated:

i. Use Image segmentation method to increase the number of samples. The size of the sample pictures in the BreaKHis data set is 700x460, and each picture is cut into 6 pictures of 230x230. The data set size is increased to 6 times. After using the segmentation method to expand the data set, the distribution of various samples is shown in Table 2.

ii. Balance the data set. It can be seen from Table 1 that among the eight types of samples in the data set, the number of samples of each type is very different, and the number of DC samples with the largest proportion is the largest. Therefore, using DC as a reference, the random non-repeated cutting method is used to expand the number of other samples. With this method, the number of samples of each classification is close to the number of DC samples. This not only balances the number of samples in the data set, but also further expands the data set. The distribution of various samples after the data set is balanced is shown in Table 3.

iii. The data set is divided into training set, validation set and test set. The sample distribution ratio is 8:1:1. The three-fold cross-validation training method is adopted to train the model three times, and the experimental results are averaged.

iv. Based on the balanced data set and the unbalanced data set, experiments are carried out separately to verify whether the balanced data set can improve the classification performance of the model.

## Proposed model

Convolutional neural network models in the field of computer vision emerge endlessly. From LeNet [19] in 1998 to MobileNet [20] proposed by Google in 2017, the application of deep

**Table 3. The distribution of various samples after the data set is balanced.**

| Magnification | Benign | | | | Benign | Malignant | | | | Malignant | Total |
|---|---|---|---|---|---|---|---|---|---|---|---|
| | A | F | PT | TA | Total | DC | LC | MC | PC | Total | |
| 40X | 4984 | 5018 | 5154 | 4994 | 20150 | 5184 | 5036 | 5130 | 5070 | 20420 | 40570 |
| 100X | 5378 | 5360 | 5326 | 5400 | 21464 | 5418 | 5020 | 5332 | 5352 | 21122 | 42586 |
| 200X | 5266 | 5284 | 5348 | 5340 | 21238 | 5376 | 5328 | 5376 | 5310 | 21390 | 42628 |
| 400X | 4636 | 4622 | 4690 | 4580 | 18528 | 4728 | 4722 | 4714 | 4728 | 18892 | 37420 |
| Patients | 4 | 10 | 3 | 7 | 24 | 38 | 5 | 9 | 6 | 58 | 82 |

learning network models in image processing is getting better and better. However, the size of neural networks is getting bigger and bigger, the structure of neural networks is getting more and more complicated, and the hardware requirements for prediction and training are getting higher and higher. If these excellent classic models are used to directly train a data set in a specific field to obtain prediction results, the prediction results obtained are often not ideal due to factors such as the small scale of the data set in a specific field. Because the medical data set has the characteristics of small scale and uneven distribution of the number of samples in the data set, it takes more effort to obtain an excellent medical prediction model.

Jason Yosinski et al. [21] from Cornell University took the lead in researching the transferability of deep neural networks. The experimental results show that it is feasible to apply the knowledge or patterns learned in a certain domain or task (called the source domain) to a different but related domain or task (called the target domain). The research results show that when deep learning is applied to image processing, the features extracted by the first-layer are basically similar to Gabor filters and color blobs. The first layer has little correlation with the type of image data set. The last layer of the network is closely related to data set in use. The features extracted in the first layer are called general features, and the features extracted in the last layer are called specific features. The combination of deep learning and migration proposed in the above paper has opened up a new way for training deep models with small data sets. Limited to the time when the depth of the network is much shallower than the current dozens or even hundreds of layers of networks, so the author said "The first layer is not relevant to the specific image data set", it should be understood as "The front end of the network is not relevant to the specific image data set "(The front end of the network is not only the first layer and there may be more layers), and the author said "The last layer of the network is closely related to the data set in use. ", it should be understood as " The back end of the network is closely related to the data set in use."(The back end of the network is not only the last layer and there may be more layers).

Fig 2 shows the Pre-training+Fine-tuning method. The so-called Pre-training+Fine-tuning method (also called transfer learning method) is to use a pre-trained model based on a large-scale data set (such as ImageNet [22]), and then fine-tune certain specific layers of the pre-trained model. The fine-tuned model is trained on a small data set in the target domain. When a source domain and a target domain is similar, it is effective for solving tasks based on small data sets [23]. Based on this method, this paper designs an automatic classification model for breast pathological tissues.

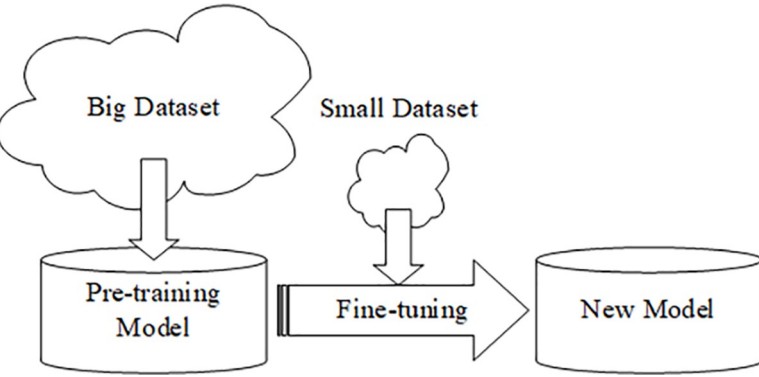

**Fig 2. Pre-training+fine-tuning method.**

**Choice of pre-trained models.** The research field of this article, namely the target domain, is image recognition of medical pathological tissues. The data sets of pathological tissue images are generally small data sets. The pre-trained models of the classic models VGG, Resnet, Xception [24], Inception [25], MobileNet [26], DenseNet [27], etc., whose source domains are all based on the large-scale dataset ImageNet. The large-scale data set ImageNet used in the source domain and the pathological tissue image data set used in the target domain seem to be irrelevant, but the target tasks of both are image classification, and the problem-solving methods are similar. Therefore, it is feasible to choose a suitable pre-trained model among these classic models. This paper chooses the pre-trained models of VGG16 and Resnet50 as the model base of the BCDnet model.

**Model design.** Based on pre-training method (transfer learning), this paper designs a parallel heterogeneous breast cancer eight-class prediction model BCDnet that adapts to small data sets such as pathological tissues. The model structure is shown in Fig 3. The detailed model structure, detailed parameter configuration and running program of the model have been uploaded to Github (Https://github.com/yeaso/bcdnet).

The notes in Fig 3:

Input (230x230x3): The input layer of the model. The data input to this layer is a 230x230 pixel color image (RGB three-channel color image).

Frozen layer: When using a pre-trained model for transfer learning on a new data set, in order to borrow the weights of certain layers of the pre-trained model, these layers of borrowed weights need to be frozen. When the model is trained on the new data set, the frozen layer is in an untrainable state, and its weight is not updated.

Resnet50.h: Pre-trained Resnet50 model weight file based on ImageNet dataset.

VGG16.h: Pre-trained VGG16 model weight file based on ImageNet data set.

W_layer: Represents the weight layers in the convolutional neural network, generally refers to the convolutional layers.

Conv_Base: Represents all convolutional layers of the pre-trained model. Here Conv_Base1 is all the convolutional layers of the pre-trained model Resnet50, and Conv_Base2 is all the convolutional layers of the pre-trained model VGG16.

Flatten: The Flatten layer is used to transition from the convolutional layer to the fully connected layer, and is used to "flatten" the input, that is, to turn multi-dimensional input features into one-dimensional features.

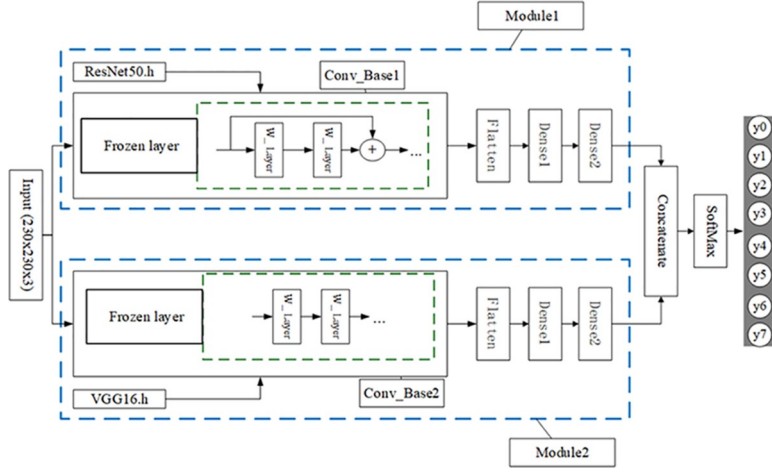

**Fig 3. The structure diagram of the model BCDnet.**

Dense: Dense layer (fully connected layer). It is generally placed at the end of the network for subsequent classification output. Each of its nodes is connected with the output node of the network front end to form a dense layer (fully connected layer).

Module: A module is composed of convolutional base and dense layer. Independently realize the information input, the information processing of the convolutional neural network, and the output of the characteristic information, and provide input information for the subsequent fusion and classification. There are two modules, Module1 and Module2, which are the main components of the model BCDnet.

Concatenate: feature fusion layer [28]. Here is the fusion of the output information of the two modules Module1 and Module2.

SoftMax: SoftMax classifier. The most used classifier in the neural network era [29].

BCDnet is composed of input layer, two parallel modules (Module1 and Module2), feature fusion layer concatenate, SoftMax classification layer and output layer.

The input layer inputs the color pathological tissue image to be trained, the shape is 230x230x3 (height x width x number of channels).

Module1 is composed of Conv_Base1, Flatten layer and two dense layers. Conv_Base1 is transformed from the pre-trained Resnet50 model. ResNet (Residual Neural Network) was proposed by Kaiming He of Microsoft Research. Based on the ImageNet dataset, Resnet50, a deep neural network with a depth of 50 layers, has achieved success in learning by using residual units. ResNet won the championship in the 2015 ILSVRC (ImageNet Large Scale Visual Recognition Challenge). Using Resnet's skip connection residual structure can make the network reach a very deep level, and has a higher performance. ResNet's contribution is mainly to discover the network "degeneration" phenomenon in the training process of deep networks, and solve the problem of too deep neural networks that are difficult to train. For a detailed explanation of Resnet, see reference [17]. In this paper, Resnet50, which performs well in the Resnet series, is used as the pre-trained model to transform, and the parameter weight file trained on ImageNet data set by Resnet50 is used to fine-tune the model.

The fine-tuning process of Conv_Base1: We know that Resnet50 has 5 residual modules Res1-Res5, and Resnet50 includes a total of 49 convolutional layers and 1 fully connected layer. First, load the Resnet50.h file based on the ImageNet data set, then freeze the loaded weights of all convolutional layers of Res1, Res2, and Res3, and unfreeze all the convolutional layers of Res4 and Res5 into a trainable state. The dense layer and classification output layer of Resnet50 are not used.

Why did this article choose to unfreeze all convolutional layers of Res4 and Res5? As mentioned above, the output features of the bottom layers in the convolutional base (such as the output features of Res1 and Res2) are more versatile and reusable, and the top layers (such as the output features of Res4 and Res5) have more specific features. Fine-tune these convolutional layers close to the top of the convolutional base so that they can be trained on the new data set, reuse the low-level features (common features) and train high-level features (specific features) to achieve the effect of transfer learning. This article chooses to unfreeze Res4 and Res5 is the optimal scheme that has been obtained after many experiments. Readers can try other freezing methods to observe the results of the model experiment, such as unfreeze Res4 or Res5, or all not unfreeze. For the specific implementation code, please refer to the file that has been uploaded to Github.

Module2 consists of Conv_Base2, flatten layer and two dense layers. Conv_Base2 is transformed from the VGG16 pre-trained model. VGG is a deep convolutional neural network jointly developed by researchers from the Computer Vision Group of Oxford University and Google DeepMind. VGG won the runner-up in the 2014 ILSVRC competition. VGG mainly explores the relationship between the depth of the convolutional neural network and its

performance. By using repeatedly stacked 3*3 small convolution kernels and using 2*2 maximum pooling layers, VGG successfully built a 16–19 layer deep convolutional neural network. The generalization ability of VGG is very good, the correct classification is also excellent, and it has good performance on different image data sets. Detailed explanation of VGG can be found in paper [16]. In this paper, VGG16, which has excellent performance in the VGG series, is used as a pre-trained model for transformation, and the parameter weight file trained on the ImageNet data set by VGG16 is used to fine-tune the model.

The fine-tuning process of Conv_Base2: VGG16 has 5 compound convolution modules block1-block5. VGG16 includes a total of 13 convolutional layers and 3 fully connected layers (dense layers). First load the VGG16.h file based on the ImageNet data set, then freeze the weights loaded by all convolutional layers of block1-block4, and unfreeze all the convolutional layers of block5 into a trainable state. The fully connected layer and classification output layer of VGG16 are not used. For the specific implementation code, please refer to the file uploaded to Github (https://github.com/yeaso/bcdnet).

The back end of Module1 and Module2 adopts a conventional design: after the convolutional layer, the multi-dimensional input is one-dimensionalized through the flatten layer and then connected to the dense layer. In order to speed up the convergence speed of model training, two dense layers (dense1 and dense2) are used consecutively here [30].

Finally, the output information of the two modules is fused in the concatenate layer and then input to the SoftMax classification layer, and finally the prediction result is output.

## Analysis of the model

On the whole, the model BCDnet is an integrated model of two pre-trained models Resnet50 and VGG16. Relevant paper points out that if the integration method is to be effective, the models involved in the integration should be as good as possible and as different as possible [23]. Resnet50 and VGG15 are models that perform well in computer vision classification and have different network structures. Therefore, this paper chooses these two models to build a parallel heterogeneous classification model. It can be seen from Fig 3 that the core components of the model BCDnet are two convolutional bases: Conv_Base1 of Module1 and Conv_Base2 of Module2. The two convolutional bases have different network structures, as shown in Figs 4 and 5. Fig 4 is the network unit structure of Conv_Base1, which is the residual convolution structure; Fig 5 is the network unit structure of Conv_Base2, which is the standard convolution structure.

In Figs 4 and 5, X represents the feature input, and W represents the weight of the convolutional layer. After each convolutional layer (W_Layer), an activation function (Relu) is applied. The output characteristic value Y of the network unit of Conv_Base1 is calculated by formula (1), and the output characteristic value Y of the network unit of Conv_Base2 is calculated by formula (2), where σ represents the nonlinear activation function Relu.

$$y = F(X, \{W_i\}) + X \qquad (1)$$

$$y = W_2\sigma(W_1X) \qquad (2)$$

The fusion of two different network structures can broaden the perspective of the model, improve the ability to obtain object features from different angles, and further improve the performance of the model. Therefore, BCDnet has the characteristics of strong combination, parallel and heterogeneous.

This model is not a simple combination of the output results of the two modules, but after the information output by the two modules is fused, the information output by the fusion layer shares the SoftMax classification function and cross entropy function. The weight of the

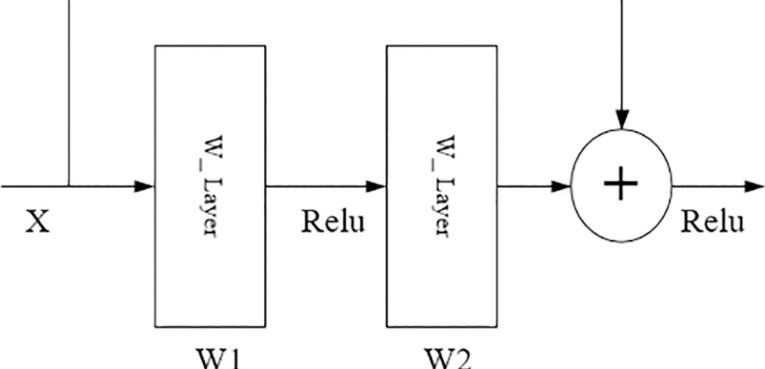

**Fig 4. Conv_Base1.**

model is updated from the joint action of the two modules. The loss cost of the overall model is represented by formula (3).

$$L = L_c(D_s, Y_s) + \lambda L_t(D_s, D_t) \tag{3}$$

L represents the final loss of the network, $L_c(D_s, Y_s)$ represents the conventional classification loss of the network in the source domain, $L_t(D_s, D_t)$ represents the loss of the fusion network to adapt to the target domain, where $\lambda$ is a weight parameter that weighs the two parts.

The output of the model prediction function softmax is based on formula (4), where $x$ represents the input sample, $y$ represents the predicted value, $x^T w_i$ represents the value of the i-th category predicted, and $K$ is the number of categories (the BCDnet model has eight categories, $K = 8$).

$$P(y = i|x) = \frac{e^{x^T w_i}}{\sum_{k=1}^{K} e^{x^T w_k}} \tag{4}$$

**Strategies for training the model.**    The defined eight-class classification labels are shown in Table 4.

In order to compare the performance of the models, based on the expanded data set in Table 2 and the balanced sample data set in Table 3, Resnet50 original model, VGG16 original

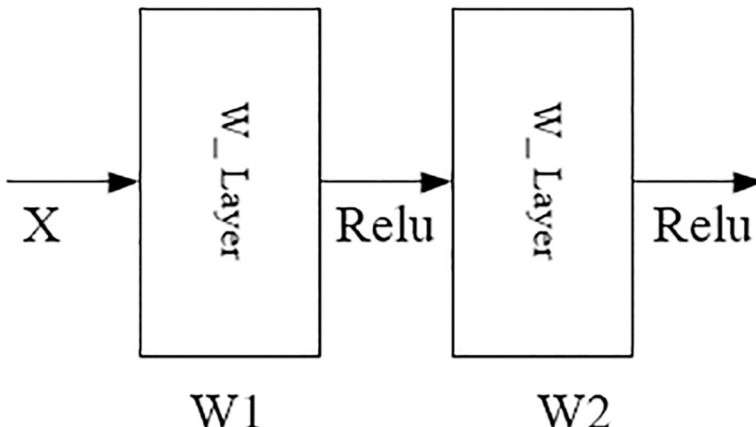

**Fig 5. Conv_Base2.**

**Table 4. Defined eight-category labels.**

| Eight-class | A | F | PT | TA | DC | LC | MC | PC |
|---|---|---|---|---|---|---|---|---|
| Labels | 01 | 02 | 03 | 04 | 11 | 12 | 13 | 14 |

model, Resnet50 pre-trained model, VGG16 pre-trained model, Resnet50 pre-trained + fine-tuning (The fine-tuning setting is the same as Module1), VGG16 pre-trained + fine-tuning (the fine-tuning setting is the same as Module2), and the BCDnet model are trained and tested separately.

Specific training strategies for the three modes of Resnet50 and VGG16:

i. Original model mode: The model does not load pre-trained weights, and the softmax output layer is set to eight-class classification output.

ii. Pre-trained model mode: The model loads pre-trained weights (for example, loads Resnet50.h or VGG16.h weight files), freezes the weights of all convolutional layers, and sets the softmax output layer to eight-class classification output.

iii. Pre-trained model + fine-tuning mode: the model loads the pre-trained weights, freezes the weights of the initial convolutional layers, trains the back-end convolutional layers and the fully connected layers (the fine-tuning settings of Resnet50 is the same as Module1, and the fine-tuning settings of VGG16 is the same as Module2), Set the softmax output layer to eight-class classification output.

**Model evaluation indicators.** Recording the results of model training and testing is the basis for further improving the model and evaluating the quality of the model. The experimental models in this article use commonly used classification model evaluation indicators: Precision and Accuracy.

Suppose true positives (positive class is judged as positive class) are denoted as TP, and false positives (negative class is judged as positive class) are denoted as FP,

False negatives (positive type is judged as negative type) are denoted as FN, true negatives (negative type is judged as negative type is denoted as TN, then the formulas of precision and accuracy are expressed as follows:

$$Precision = \frac{TP}{TP + FP} \tag{5}$$

$$Accuracy = \frac{TP + TN}{TP + FP + TN + FN} \tag{6}$$

## Results

According to the aforementioned training strategy, based on eight data sets of 40X, 40X+, 100X, 100X+, 200X, 200X+, 400X, 400X+, Resnet50 original model, VGG16 original model, Resnet50 pre-trained model, VGG16 pre-trained model, Resnet50 pre-trained model + fine-tuning (the fine-tuning setting is the same as Module1), the VGG16 pre-trained model + fine-tuning (the fine-tuning setting is the same as Module2) and the BCDnet model are trained and tested separately. As shown in Figs 6 and 7, they are record curves of training accuracy, verification accuracy, training loss, and verification loss in the training process of BCDnet based on 100X and 100X+ data sets. Fig 6 is the recording curve of BCDnet based on the 100X data set, and Fig 7 is the recording curve of BCDnet based on the 100X+ data set. In the legend, Training acc means training accuracy, and Validation acc means validation accuracy.

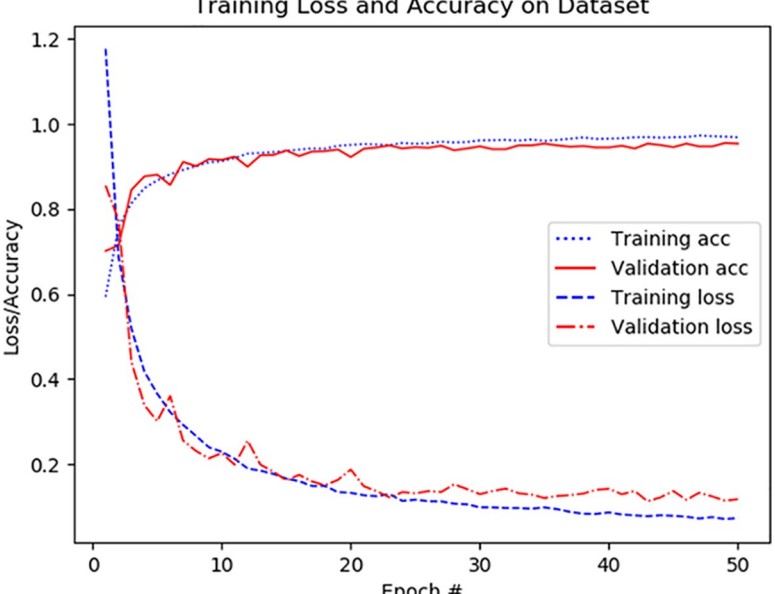

**Fig 6. Recording curve of training process based on 100X data set.**

In order to facilitate the observation of the experimental performance of each model, the experimental results are represented by a histogram, as shown in Fig 8.

In the legend, R50 represents the original model of Resnet50, R50p represents the pre-trained Resnet50 model based on the ImageNet data set, R50pt represents the pre-trained Resnet50 model based on the ImageNet data set + fine-tuning, V16 represents the original model of VGG16, V16p represents the VGG16 model pre-trained based on the ImageNet data set and V16pt represents the pre-trained VGG16 model based on the ImageNet data set + fine-tuning.

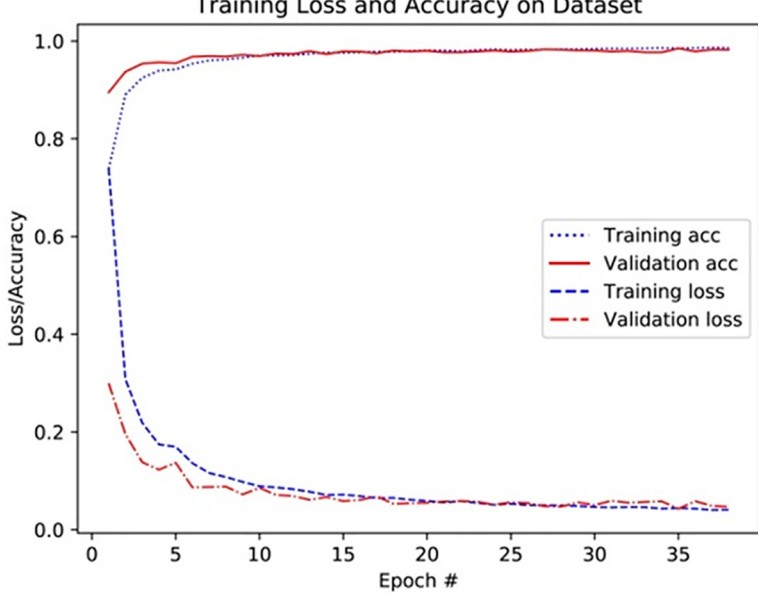

**Fig 7. Recording curve of training process based on 100X+ data set.**

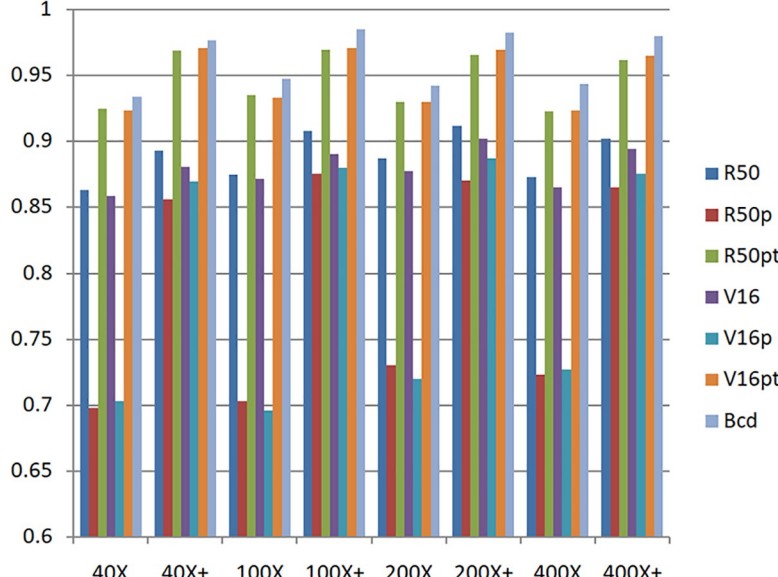

**Fig 8. Experimental results based on unbalanced data set and balanced data set.**

Based on the unbalanced data set and the balanced data set, the experimental results of the eight-class classification precision of the models of BCDnet, R50pt, and V6pt are represented by a histogram, as shown in Figs 9 and 10. The data shown in Fig 9 is based on the unbalanced data set (40X). The data shown in Fig 10 is based on the balanced data set (40X+). The results of other corresponding groups are similar to this, and are omitted here.

## Analysis and discussion

It can be seen from the experimental results in Fig 8 that with a balanced data set, the performance of each model is significantly improved. The performance of the original model is better than the pre-trained model. The performance of the pre-trained model + fine-tuning is significantly better than the original model. The BCDnet integrated with the pre-trained model + fine-tuning method performed best. Obviously, BCDnet takes advantage of the cooperation of multiple models. For the result that the original model performs better than the pre-trained model, this article believes that the pre-trained model only uses the loading weights

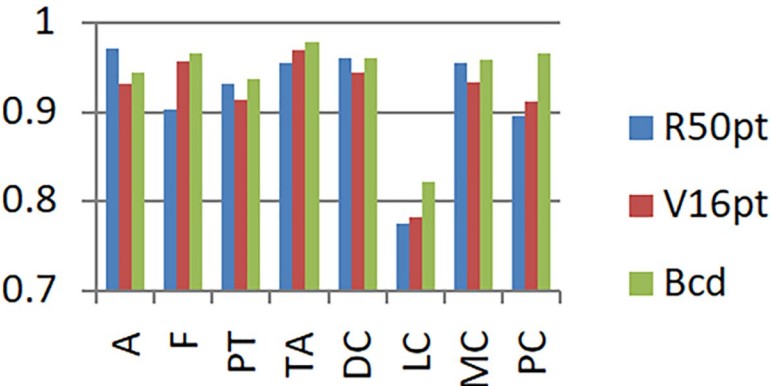

**Fig 9. Experimental results of eight-class classification precision based on 40X data sets.**

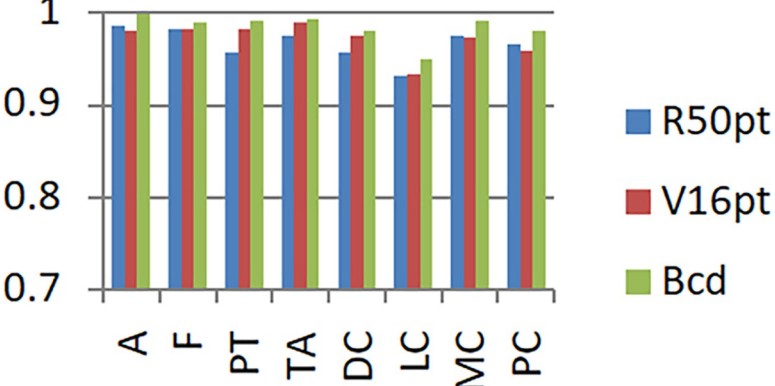

**Fig 10. Experimental results of eight-class classification precision based on 40X+ data sets.**

and freezes all the convolutional layer weights (when training on the new data set, the loaded weights are not updated), due to the loaded weights from the ImageNet data set, the samples in the ImageNet data set are very different from the samples in the pathological tissue set, and the frozen layers do not participate in the training on the pathological tissue data set, so the performance of the pre-trained model is poor.

The reason why the pre-trained model + fine-tuning performs significantly better than the original model is that the model reused the weights of the bottom-level features from the pre-trained model by the large dataset ImageNet, and changed the weights of the top-level features by the breast cancer pathological tissue dataset.

In view of the outstanding advantages of the pre-trained model + fine-tuning mode, the following only introduces the Resnet50 pre-trained model + fine-tuning (denoted as R50pt), VGG16 pre-trained model + fine-tuning (denoted as V16pt) and the integrated model BCDnet (denoted as BCD) Experimental results.

Figs 11 and 12 show the accuracy comparison of R50pt, V16pt and BCD based on the unbalanced data sets (40X, 100X, 200X, 400X) and the balanced data sets(40X+, 100X+, 200X+, 400X+).

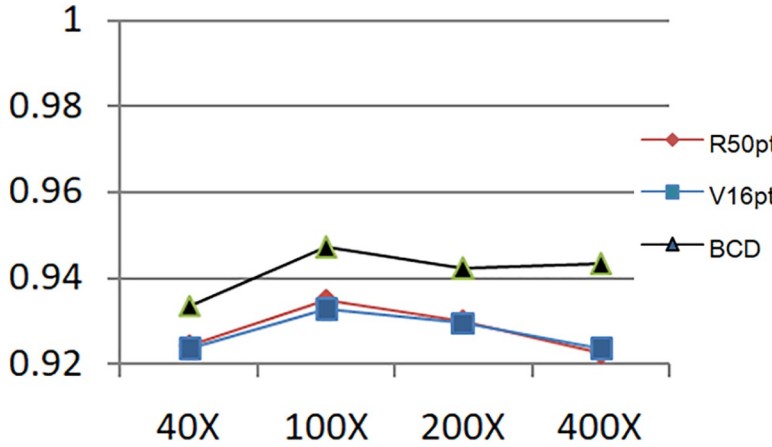

**Fig 11. Comparison of accuracy of R50pt, V16pt and BCD based on unbalanced data sets.**

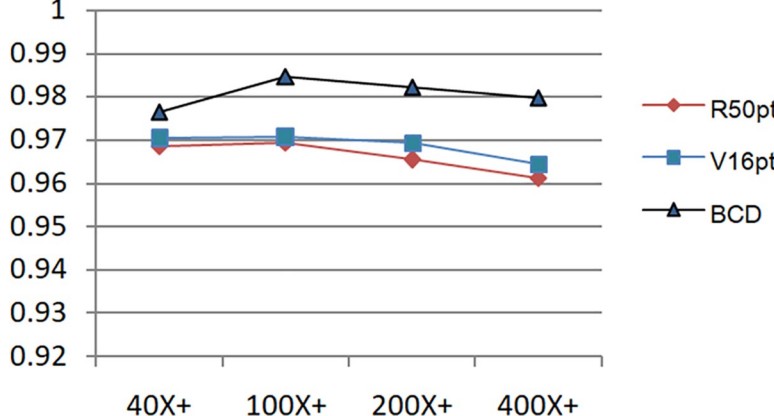

**Fig 12. Comparison of accuracy of R50pt, V16pt and BCD based on balanced data sets.**

Judging from the experimental results in Fig 8 and the comparison curve in Figs 11 and 12, the experimental results have the following characteristics:

i. Based on the experimental results of sample data at four magnifications of 40X, 100X, 200X, and 400X, the overall difference is not big, and the performance of 100X, 200X, 100X+, and 200X+ is slightly better. The reason should be related to the larger proportion of their samples in the data set, and positive correlation with the data distribution in Table 1.

ii. Balanced data sets (with a + sign, such as Data200X+) are compared with unbalanced data sets (such as Data200X). Balanced data sets can significantly improve the performance of the model. The number of samples for each classification of the balanced data set increases, and the number of samples for each classification is relatively close, so the model performance improvement is reasonable.

iii. The experimental results of R50pt, V16pt and BCDnet have similar performance distributions in all data groups. BCDnet model is better than R50pt and V16pt. BCDnet has the characteristics of strong combination and selection of the best. It shows that the design of BCDnet model, the choice of basemodle and the strategy of fine-tuning are reasonable.

iv. Observe Figs 11 and 12. It can be seen from Fig 11 that based on the unbalanced data sets, the experimental results of BCDnet exceed R50pt and V16pt by about two points. It can be seen from Fig 12 that based on the balanced data sets, the experimental results of BCDnet exceed R50pt and V16pt by about one point, indicating that the BCDnet model has better adaptability and tolerance to small data sets. Of course, because the experimental results based on the balanced data set have reached more than 96%, BCDnet exceeds R50pt and V16pt by one point, which is also great.

v. In order to facilitate the analysis of the precision of each class in the eight-class classification of the model, the curves shown in Figs 13 and 14 are drawn according to Figs 9 and 10. (Here, take the precision curve of eight-class classification based on 40X and 40X+ data sets as an example. The trends of other groups are also similar to Figs 13 and 14, so not repeating them here.).

It can be seen from Figs 13 and 14 that the model has the lowest accuracy in identifying lobular carcinoma (LC). Based on the unbalanced data, the model's accuracy for identifying lobular carcinoma is about 80%. Based on the balanced data, the accuracy of the model to identify

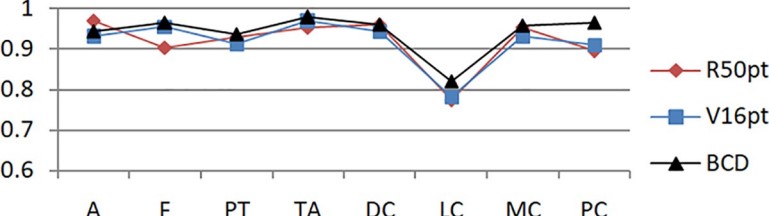

**Fig 13. Precision curve of eight classifications based on 40X.**

LC is close to 95%. It shows that the use of expanded data sets in multi-classification and keeping the number of various types of samples balanced has a significant effect on the identification of samples that are difficult to cluster.

It is also found from Figs 13 and 14 that even if the data set equalization is not performed, some types of pathological tissue images can be easily recognized correctly, and even if the data set equalization is performed, there will always be some pathological tissue images that cannot be correctly recognized. According to the definition of precision in formula (5), the reason why the recognition accuracy of a certain sample A is low is because there are more samples of other categories that are mistakenly classified into the category of sample A. It can also be understood in this way that the A sample is "unclear and not unique" and has certain attributes of other samples. This article refers to this type of sample as having weak characteristics. Therefore, in multi-class classification, in addition to using expanded sample data and balanced sample data to improve the performance of the model, further theoretical research and technical method improvement should be carried out for the extraction and optimization of weak feature samples. Solving the problem of weak features is sometimes more effective than improving the model [31].

## Conclusion

Aiming at the characteristics of complex and diverse biopathological tissue images and small medical data sets, this paper uses pre-training + fine-tuning and model integration techniques to design a breast cancer diagnosis model BCDnet. The experimental results prove that the method of expanding the data set and balancing the number of classified samples can further improve the performance of the model. The experimental results prove that the model designed in this paper runs stably, and the correct recognition rate of the model is higher than 98%. Although the method in this paper is based on breast cancer pathological tissue images, it is also suitable for multi-class visual recognition of other pathological tissue images. Although methods such as parameter optimization and data set improvement have been adopted, the model has achieved high performance, but there are still some samples that are always difficult to identify. For example, the recognition accuracy of lobular carcinoma (LC) in this article is

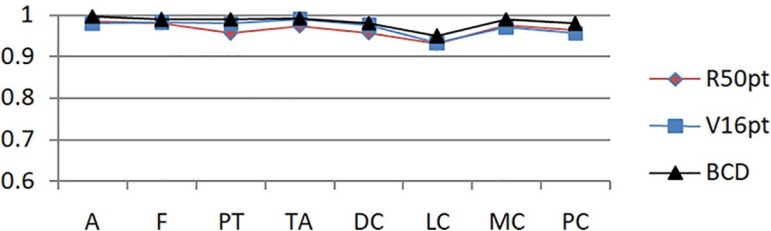

**Fig 14. Precision curve of eight classifications based on 40X+.**

about 95%. Although this value looks good, it lags behind the performance of the overall model. Therefore, in the future, we should focus on studying these weak feature samples that are difficult to cluster and train, and explore methods of extracting and optimizing weak feature samples to further improve the performance of the model.

## Author Contributions

**Data curation:** Qingfang He.

**Formal analysis:** Qingfang He.

**Funding acquisition:** Guang Cheng.

**Investigation:** Guang Cheng, Huimin Ju.

**Methodology:** Qingfang He.

**Project administration:** Guang Cheng.

**Software:** Qingfang He, Huimin Ju.

**Supervision:** Guang Cheng.

**Validation:** Qingfang He.

**Writing – original draft:** Qingfang He.

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
