## [Decision Letter · Decision Letter 0]

24 Feb 2021

PONE-D-20-37399

BCDnet: Parallel Heterogeneous Eight-category Automatic Diagnosis Model of Breast Pathology

PLOS ONE

Dear Dr. He,

Thank you for submitting your manuscript to PLOS ONE. After careful consideration, we feel that it has merit but does not fully meet PLOS ONE’s publication criteria as it currently stands. Therefore, we invite you to submit a revised version of the manuscript that addresses the points raised during the review process.

As you will infer from below that there was a disagreement among the reviewers regarding enthusiasm for this work. Reviewer 1 was of the view that manuscript partly describes a technically sound piece of scientific research and recommended major revision. However,  Reviewer 2 was of the view that your work did not describe technically sound piece of scientific research and recommended reject. 

After thorough consideration of comments of reviewers, my decision is "major revision". Please incorporate comments raised by both reviewers. 

We look forward to receiving your revised manuscript.

Kind regards,

Gulistan Raja

Academic Editor

PLOS ONE

Journal Requirements:

2.  We note that Figures 4 and 6 in your submission contain copyrighted images. All PLOS content is published under the Creative Commons Attribution License (CC BY 4.0), which means that the manuscript, images, and Supporting Information files will be freely available online, and any third party is permitted to access, download, copy, distribute, and use these materials in any way, even commercially, with proper attribution. For more information, see our copyright guidelines: http://journals.plos.org/plosone/s/licenses-and-copyright.

2.1.        You may seek permission from the original copyright holder of Figures 4 and 6 to publish the content specifically under the CC BY 4.0 license.

2.2.    If you are unable to obtain permission from the original copyright holder to publish these figures under the CC BY 4.0 license or if the copyright holder’s requirements are incompatible with the CC BY 4.0 license, please either i) remove the figure or ii) supply a replacement figure that complies with the CC BY 4.0 license. Please check copyright information on all replacement figures and update the figure caption with source information. If applicable, please specify in the figure caption text when a figure is similar but not identical to the original image and is therefore for illustrative purposes only.

Reviewers' comments:

Reviewer's Responses to Questions

**Comments to the Author**

1. Is the manuscript technically sound, and do the data support the conclusions?

Reviewer #1: Partly

Reviewer #2: No

2. Has the statistical analysis been performed appropriately and rigorously? 

Reviewer #1: No

Reviewer #2: No

3. Have the authors made all data underlying the findings in their manuscript fully available?

Reviewer #1: Yes

Reviewer #2: Yes

4. Is the manuscript presented in an intelligible fashion and written in standard English?

Reviewer #1: No

Reviewer #2: No

5. Review Comments to the Author

Reviewer #1: 1) The overall grammar and sentence framing in the paper are very weak, making the article quite unreadable. Grammar checks were not done at all.

2) The author should try to give data of the study region too, in the Introduction.

3) Example:

line 83… "Most of the research focuses on 84 relatively simple two classification"…. Here it appears that the author's research focuses on this.

Line 85… "…moving from two classifications….." maybe binary is a better term

Line 87…. "Improving the accuracy of the model by improving the data set has also been paid more and more attention". How has the data set been improved? Is this sentence part of literature, or this work?... is unclear.

Line 165… "If you load the weights and freeze all the convolutional layer weights, because the loaded weights come from the training of the ImageNet image set, the samples in the ImageNet image set have little correlation with the samples in the pathological tissue image set, so the experimental results of strategy 3 are poor." Very bad sentence framing.

Line 302… "…sample number distribution…." What does this mean?

4) Line 103…"….using random non-repetitive segmentation of the original image sample to enhance the data for the small sample can significantly improve the model performance". This sentence must have the support of citation.

5) Line 141… "Choose pre-trained models" Title should be appropriately named. Maybe Choice of pre-trained models.

6) It is not understood how Figure 2 describes the model structure. Further, no description is added of the labels in the figure.

7) Table 2 mentions the raw dataset numbers. There is no breakup of what the enhanced dataset numbers are.

8) Line 295… "…The abscissa is the xx patien…” What is this xx?

9) The paper has commas and blanks spaced out at will. Seems like a very hurriedly written paper.

10) The paper's only novelty is the 8-class classification, which is interesting, and hence carries a chance for re-submission.

11) But, if a new AI method/theory is being reported in the paper, it should be compared and validated against at least one other common data set (bench mark datasets) for which a published study exists using at least one other method/approach, and preferably a method/approach which has been widely used in the field.

12) Also, baselines are not strong (e.g., did not compare with state-of-the-art approaches)

Reviewer #2: In this manuscript, the authors clearly define their problem – there are tons of studies that do benign/normal binary classification but few that address the need to be able to distinguish between different kinds of lesions.

They fuse two pretrained CNNs by concatenating their output features and retrain a subset of layers in order to classify small histopathological images into one of eight categories. They test the performance of their model at different magnifications and with/without using data augmentation.

Overall, the manuscript is difficult to follow mainly to do deviance from structural conventions and lack of organization. There are also experimental details which are unclear or left out which yields the work unreproducible.

My comments are below.

Major

Generally, research articles are structured with abstract, introduction, methods, results, discussions, and conclusion headings. Can the authors reorganize their manuscript according to these conventions?

Though some ideas presented by the authors can be understood, some paragraphs are difficult to understand. For example, this reviewer finds it difficult to understand what is being said on lines 101-109. There are also numerous instances of sentence fragments, run-on sentences, comma splices, etc. Please have someone who is fluent in English to proof read the manuscript.

It is not clear in the "Choose pre-trained models" subsection how models are trained nor what results are reported in Table 1. The authors state that they are using strategies 2, 3, and 4 to pre-train on a small set of pathological tissues. What performance metric is being measured? Accuracy? How are large tissue images being processed by the fixed-input CNNs? Can the authors describe the distribution of the dataset before presenting results?

The authors do well describing Resnet50 and VGG16. This reviewer can understand that the 30th and 10th layers respectively are frozen as well as preceding layers. Then, deep layers are retrained. However, it is not clear what parameters are utilized to fine-tune these networks (learning rate, optimizer, weight biases, etc.). Furthermore, it is not clear what is the 30th layer or 10th layer of these networks. Generally when describing CNN layers, we prefer to refer to abstracted components of the network. For example, Resnet50 has five residual "blocks." It would help to clarify these sections.

The resampling strategy proposed by the authors to combat class imbalances is well-founded. However, the description provided is not clear enough to reproduce the resampling strategy. Can the authors rephrase their description?

The authors make it a point that class imbalance will be an issue, but given the results, it seems as if it hardly makes a difference. Yes, the majority of performance metrics are higher with augmentation but only a little bit. Can the authors provide some discussion on this?

Minor

It is a bit unusual to compare lung and breast cancer as a race to first place on lines 40-41 on page 2. Please rephrase.

The transition from pathological diagnosis to image classification on lines 46-47 on page 2 is non-sequitur. Please rephrase.

What does "pre-screened" mean on line 233 page 7?

What is meant by "mark" in Table 3 (page 10)?

What is verification accuracy and loss?

Can the authors represent Table 4 as a grouped bar graph? It is difficult to see the differences in performance with such a large table.

One of the columns for Dataset400X in Table 4 contain results rather than marks. Please fix.

Does Table 4 report cross-validation metrics?

6. PLOS authors have the option to publish the peer review history of their article (what does this mean?). If published, this will include your full peer review and any attached files.

Reviewer #1: No

Reviewer #2: No

---

## [Author Response · Author response to Decision Letter 0]

9 Apr 2021

Dear editor and two reviewers:

First of all, thank you for your comments and suggestions on this article, and sincerely thank you for your guidance and help. The two reviewers showed a high level of professionalism and reviewed this article very carefully. The reviewers’ comments and suggestions are very pertinent. We have learned a lot from these opinions. In response to your comments and suggestions, we have carefully revised the article based on the comments of the two reviewers.

---

## [Decision Letter · Decision Letter 1]

3 Jun 2021

PONE-D-20-37399R1

BCDnet: Parallel heterogeneous eight-class classification model of breast pathology

PLOS ONE

Dear Dr. He,

Thank you for submitting your manuscript to PLOS ONE. After careful consideration, we feel that it has merit but does not fully meet PLOS ONE’s publication criteria as it currently stands. Therefore, we invite you to submit a revised version of the manuscript that addresses the points raised during the review process.

Specifically, the English language should be improved further.

We look forward to receiving your revised manuscript.

Kind regards,

Gulistan Raja

Academic Editor

PLOS ONE

Journal Requirements:

Additional Editor Comments (if provided): English language should be improved further

Reviewers' comments:

Reviewer's Responses to Questions

**Comments to the Author**

1. If the authors have adequately addressed your comments raised in a previous round of review and you feel that this manuscript is now acceptable for publication, you may indicate that here to bypass the “Comments to the Author” section, enter your conflict of interest statement in the “Confidential to Editor” section, and submit your "Accept" recommendation.

Reviewer #2: All comments have been addressed

Reviewer #3: All comments have been addressed

Reviewer #4: All comments have been addressed

2. Is the manuscript technically sound, and do the data support the conclusions?

Reviewer #2: Yes

Reviewer #3: No

Reviewer #4: Yes

3. Has the statistical analysis been performed appropriately and rigorously? 

Reviewer #2: N/A

Reviewer #3: No

Reviewer #4: N/A

4. Have the authors made all data underlying the findings in their manuscript fully available?

Reviewer #2: Yes

Reviewer #3: Yes

Reviewer #4: Yes

5. Is the manuscript presented in an intelligible fashion and written in standard English?

Reviewer #2: Yes

Reviewer #3: No

Reviewer #4: Yes

6. Review Comments to the Author

Reviewer #2: The authors have done a very good job while revising the manuscript; however, a couple of my comments were overlooked or not properly addressed. Looking at the amount of effort the authors have put in revising the manuscript, I recommend this publication for publication.

Reviewer #3: This is the review of the paper titled " BCDnet: Parallel heterogeneous eight-class classification model of breast pathology"

The paper has many flaws which I will list some of them.

1- poor novelty, there is nothing new in the paper.

2- poor presentation, many sentences are ambiguous.

3- the used dataset is old and very high results achieved on the dataset.

4- no confusion matrics are presented and visualization of the learned filters.

5- Poor English

This paper is not in the level of PLOS One journal, I would suggest submitting it to a different journal.

Reviewer #4: This paper can be accepted since the authors have revised it according to the comments of the reviewers.

7. PLOS authors have the option to publish the peer review history of their article (what does this mean?). If published, this will include your full peer review and any attached files.

Reviewer #2: No

Reviewer #3: No

Reviewer #4: No

---

## [Author Response · Author response to Decision Letter 1]

8 Jun 2021

Dear editor and reviewers: 

First of all, thanks for your comments and suggestions on this article, and sincere thanks for your guidance and help. 

Two reviewers (Reviewer #2 and Reviewer #4) showed a high level of professionalism and gave positive opinions on the revised manuscript of this article. 

Reviewer 3# gave many negative opinions, for which we make the following response.

“ 2. Is the manuscript technically sound, and do the data support the conclusions?

Reviewer #3: No”

Reply: Based on deep convolutional neural network, model fusion, and feature fusion technology, this paper proposes an eight-class classification of breast pathological tissue diagnosis model (BCDnet). Experiments were carried out on the public BreaKHis scientific research data set, and the experimental results proved that BCDnet achieved eight-class classification of breast cancer pathological tissues, and the classification accuracy was higher than the published research results.

“3. Has the statistical analysis been performed appropriately and rigorously?

Reviewer #3: No”

Reply: The article provides detailed data records, data grouping records and experimental results analysis. It also provides a comparative analysis of other models and the BCDnet model. Such as Table 1 Overview of BreaKHis data set, Table 2 The distribution of various samples after the image segmentation method is used to expand the data set, Table 3 The distribution of various samples after the data set is balanced, Fig 5. Recording curve of training process based on 100X, 100X+ data set, Fig 6 Experimental results based on unbalanced data set and balanced data set, Fig 7. Experimental results of eight-class classification precision based on 40X and 40X+ data sets, Fig 8. Comparison of accuracy of R50pt, V16pt and BCD based on unbalanced data sets and balanced data sets and Fig 9. Precision curve of eight classifications based on 40X and 40X+ , etc.

“5. Is the manuscript presented in an intelligible fashion and written in standard English?

Reviewer #3: No” 

Reply: We do have shortcomings in English, and we need to improve our language skills. We had repeatedly revised the original revised manuscript, trying to express the content clearly. In this revised manuscript, we have looked for errors in language expression and revised them, and we also invited English professionals to review them to further improve our English expression skills.

“Reviewer #3: This is the review of the paper titled " BCDnet: Parallel heterogeneous eight-class classification model of breast pathology"

The paper has many flaws which I will list some of them.

1- poor novelty, there is nothing new in the paper.

2- poor presentation, many sentences are ambiguous.

3- the used dataset is old and very high results achieved on the dataset.

4- no confusion matrics are presented and visualization of the learned filters.

5- Poor English”

About 1- poor novelty, there is nothing new in the paper. 

Reply: We use advanced technology in deep learning (model fusion, etc.) to independently design BCDnet to solve the problem of low multi-class recognition rate of breast cancer pathological tissues. Judging from the results published so far, our model and method are leading.

About 2- poor presentation, many sentences are ambiguous. 

Reply: The original revised manuscript has been repeatedly revised. In this revision, the language has continued to be improved.

About 3- the used dataset is old and very high results achieved on the dataset. 

Reply: Since 2015, Spanhol et al. published the BreaKHis breast cancer pathology image data set [10]. Based on this data set, a series of research results have been achieved in breast cancer recognition using convolutional neural networks. The data set used in this article (BreaKHis data set) is a public scientific research data set, and now it is widely used to train and test breast cancer diagnostic models.

Our model adopts the current popular model fusion and other technologies, and adopts improved experimental methods according to the characteristics of pathological tissues. Therefore, the experimental results reach a good level. The experimental model, operating procedure and data set of this article have been published, and we hope that everyone will evaluate to further improve the experimental results.

About 4- no confusion matrics are presented and visualization of the learned filters. 

Reply: Confusion matrix is one of the methods to evaluate the model, which has the characteristics of intuitive expression of precision and accuracy. However, to express the same content, at least 8x8 tables are required for the eight-class classification, and the standard evaluation criteria used in this article only need 1x8 tables. There are 56(7 models, 8 data sets, 7x8=56) groups of comparative experiments in this article. If 56 confusion matrix diagrams are provided, it will take up a lot of article space and it is not convenient to compare experimental results. Therefore, this article adopts the standard evaluation criteria: Precision and Accuracy.

The visualization of the learned filters is an auxiliary tool when training the model. Observation for training from scratch can assist in adjusting the parameters of the underlying network to improve the model. This article uses the pre-trained model and model fusion technology. The visualization of the learned filters is of little significance to the model adjustment, so this article does not provide the visualization of the learned filters diagrams.

About 5- Poor English” 

Reply: The original revised manuscript has been repeatedly revised. In this revision, the language has continued to be improved.

Reply to edit: In response to the opinion given by the academic editor: "Specifically, the English language should be improved further." We have repeatedly checked the language of the article, corrected the grammatical errors, and unified the translation of professional words. and finally asked a professional thesis translation teacher to review it. 

Best regards!

---

## [Editor Report · Decision Letter 2]

14 Jun 2021

BCDnet: Parallel heterogeneous eight-class classification model of breast pathology

PONE-D-20-37399R2

Dear Dr. He,

We’re pleased to inform you that your manuscript has been judged scientifically suitable for publication and will be formally accepted for publication once it meets all outstanding technical requirements.

Kind regards,

Gulistan Raja

Academic Editor

PLOS ONE
---

## [Editor Report · Acceptance letter]

2 Jul 2021

PONE-D-20-37399R2 

BCDnet: Parallel heterogeneous eight-class classification model of breast pathology 

Dear Dr. He:

I'm pleased to inform you that your manuscript has been deemed suitable for publication in PLOS ONE. Congratulations! Your manuscript is now with our production department. 

Kind regards, 

on behalf of

Dr. Gulistan Raja 

Academic Editor

PLOS ONE